# 5-Methylcytosine and 5-Hydroxymethylcytosine in Scrapie-Infected Sheep and Mouse Brain Tissues

**DOI:** 10.3390/ijms24021621

**Published:** 2023-01-13

**Authors:** Adelaida Hernaiz, Sara Sentre, Marina Betancor, Óscar López-Pérez, Mónica Salinas-Pena, Pilar Zaragoza, Juan José Badiola, Janne Markus Toivonen, Rosa Bolea, Inmaculada Martín-Burriel

**Affiliations:** 1Laboratorio de Genética Bioquímica (LAGENBIO), Facultad de Veterinaria, Universidad de Zaragoza, IA2, IIS Aragón, 50013 Zaragoza, Spain; 2Centro de Encefalopatías y Enfermedades Transmisibles Emergentes (CEETE), Facultad de Veterinaria, Universidad de Zaragoza, IA2, IIS Aragón, 50013 Zaragoza, Spain; 3Centro de Investigación Biomédica en Red de Enfermedades Neurodegenerativas (CIBERNED), Instituto Carlos III, 28029 Madrid, Spain

**Keywords:** 5-methylcytosine, 5-hydroxymethylcytosine, prion diseases, scrapie

## Abstract

Scrapie is a neurodegenerative disorder belonging to the group of transmissible spongiform encephalopathies or prion diseases, which are caused by an infectious isoform of the innocuous cellular prion protein (PrP^C^) known as PrP^Sc^. DNA methylation, one of the most studied epigenetic mechanisms, is essential for the proper functioning of the central nervous system. Recent findings point to possible involvement of DNA methylation in the pathogenesis of prion diseases, but there is still a lack of knowledge about the behavior of this epigenetic mechanism in such neurodegenerative disorders. Here, we evaluated by immunohistochemistry the 5-methylcytosine (5mC) and 5-hydroxymethylcytosine (5hmC) levels in sheep and mouse brain tissues infected with scrapie. Expression analysis of different gene coding for epigenetic regulatory enzymes (*DNMT1*, *DNMT3A*, *DNMT3B*, *HDAC1*, *HDAC2*, *TET1*, and *TET2*) was also carried out. A decrease in 5mC levels was observed in scrapie-affected sheep and mice compared to healthy animals, whereas 5hmC displayed opposite patterns between the two models, demonstrating a decrease in 5hmC in scrapie-infected sheep and an increase in preclinical mice. 5mC correlated with prion-related lesions in mice and sheep, but 5hmC was associated with prion lesions only in sheep. Differences in the expression changes of epigenetic regulatory genes were found between both disease models, being differentially expressed *Dnmt3b*, *Hdac1*, and *Tet1* in mice and *HDAC2* in sheep. Our results support the evidence that DNA methylation in both forms, 5mC and 5hmC, and its associated epigenetic enzymes, take part in the neurodegenerative course of prion diseases.

## 1. Introduction

Epigenetics is a field of study focused on heritable changes in gene activity or function that are not associated with any change in the DNA sequence itself [1]. One of the most studied epigenetic modifications is DNA methylation, which consists of the covalent addition of a methyl group to the C-5 position of the nucleobase cytosine to form 5-methylcytosine (5mC) [2]. In mammals, this DNA methylation process occurs predominantly at cytosine residues within CpG dinucleotides [3]. Moreover, multiple forms of DNA methylation have been identified including 5mC, its hydroxylated derivative 5-hydroxymethylcytosine (5hmC), and its ensuing oxidation products 5-formylcytosine (5fC) and 5-carboxylcytosine (5caC) [4]. Among these forms, 5mC and 5hmC are relatively stable and abundant in mammalian genomes [5], whereas 5fC and 5caC are rarer and can be transiently removed [6].

Different enzymes act as mediators of the DNA methylation process. DNA methyltransferases (DNMTs) are responsible for producing 5mC by covalently adding a methyl group at the C-5 position of cytosines. There are three members of the DNMT family that directly catalyze the addition of methyl groups onto DNA: DNMT1 (DNA methyltransferase 1), DNMT3A (DNA methyltransferase 3 alpha), and DNMT3B (DNA methyltransferase 3 beta) [1]. DNMT3A and DNMT3B are de novo methyltransferases that target cytosines of previously unmethylated CpG dinucleotides, having an equal preference for hemimethylated and unmethylated DNA, and are essential in the de novo methylation of the genome during development [7]. On the other hand, established DNA methylation patterns are stably preserved over cell divisions by DNMT1, a maintenance enzyme that preserves existing methylated sites with a preference for hemimethylated DNA [8].

DNA methylation is involved in the regulation of gene expression and the 5mC form is predominantly associated with gene silencing [9]. DNMT1, DNMT3A, and DNMT3B can repress transcription through an association with histone deacetylases HDAC1 (histone deacetylase 1) and HDAC2 (histone deacetylase 2), enzymes that remove acetyl groups from histones facilitating chromatin compaction and repressing transcription [10].

The 5hmC form results from the addition of a hydroxyl group to 5mC by the ten-eleven translocation enzymes (TETs) [11]. This TET enzyme family comprises three cytosine dioxygenases: TET1 (tet methylcytosine dioxygenase 1), TET2 (tet methylcytosine dioxygenase 2), and TET3 (tet methylcytosine dioxygenase 3) [4]. As previously mentioned, among the three oxidated forms of 5mC, 5hmC is the most stable varying its presence significantly between tissues [12]. Recent findings report that 5hmC is predominantly enriched in the vicinity of transcription factor binding sites, including distal regulatory elements and gene bodies of highly expressed genes, and is less abundant at gene promoter regions [13,14]. This distribution suggests that 5hmC may be associated with stable regulation of gene expression across the genome, potentially counteracting the gene repression produced by 5mC [4].

The regulation of the de novo methylation and demethylation is of great importance for the differentiation and maturation of the mammalian central nervous system (CNS) [1]. DNMTs act coordinated, regulating the 5mC methylation patterns of the neural populations and organizing neuronal development [15]. Moreover, DNMT1 and DNMT3A are involved in the synaptic plasticity of postmitotic neurons and play a role in learning and memory in the adult brain [16]. In addition to 5mC, 5hmC is highly enriched in the CNS in comparison to other tissues [17] and, although less is known about the functions of 5hmC in the brain, some studies suggest the involvement of 5hmC in neurodevelopment and neurological function [18,19].

Just like DNA methylation, in both 5mC and 5hmC forms, it is associated with normal development and function of the CNS; altered DNA methylation patterns have been related to a variety of neurological and neurodegenerative disorders, including Alzheimer’s (AD) [20,21] and Parkinson’s (PD) [22,23] diseases, as well as Transmissible Spongiform Encephalopathies (TSEs), or prion diseases [24,25].

Prion diseases are a group of neurodegenerative disorders affecting humans and other animals [26] that are caused by an infectious isoform of the innocuous cellular prion protein (PrP^C^), partially resistant to proteases and prone to form aggregates, called PrP^Sc^ [27]. The accumulation of the PrP^Sc^ in the CNS causes spongiform degeneration, glial cell activation, and neuronal loss [28]. Among the various types of TSEs, ovine scrapie was the first discovered and is considered a good model for the study of different aspects of prion diseases [29,30].

Although recent studies have shed light on the possible roles of DNA methylation in the pathogenesis of prion diseases [24,25,31,32,33], knowledge in this field is still scarce. Little is known about how 5mC patterns and DNA methylation enzymes behave throughout the course of these diseases and, to the best of our knowledge, no study has yet explored the role of 5hmC in prion pathology. The present study aimed to evaluate the 5mC and 5hmC brain profiles in a transgenic scrapie mouse model and sheep naturally infected with scrapie. Moreover, expression analysis of the genes coding different enzymes involved in the DNA methylation process was performed along with a correlation analysis in order to find possible associations between the brain levels of 5mC and 5hmC with the expression levels of genes encoding DNA methylation enzymes and with prion-related lesions.

## 2. Results

### 2.1. Scrapie Characterization

Figure 1 shows the PrP^Sc^ deposition and vacuolation of the analyzed brain areas in Tg338 mice. The obex, mesencephalon, thalamus, and hypothalamus were the areas with higher PrP^Sc^ deposits. Regarding vac vacuolation, obex, mesencephalon, hypothalamus, and septal area/striatum were the areas that showed higher levels of spongiosis.

The scrapie characterization of the sheep used in this study was performed previously in an earlier publication from our group [34].

### 2.2. Expression Levels of Genes Involved in Epigenetic Regulation

#### 2.2.1. Gene Expression Profile in Tg338 Mice

The expression profile of seven genes described to be involved in epigenetic regulation (*Dnmt1*, *Dnmt3a*, *Dnmt3b*, *Hdac1*, *Hdac2*, *Tet1*, and *Tet2*) was analyzed by quantitative real-time PCR (RT-qPCR) in the thalamus of Tg338 mice, a transgenic scrapie mouse model expressing a transgenic VRQ allele of the ovine *PRNP* gene under the ovine PrP promoter [35,36]. The mice were divided into four groups of study: clinical and preclinical mice infected with scrapie and their corresponding controls, designated as clinical control and preclinical control, respectively.

In all the studied genes, preclinical mice showed lower expression levels compared to their controls. This decreased expression was statistically significant in *Dnmt3b* (*p* < 0.05) and *Tet1* (*p* < 0.05), and showed a trend to signification (*p* = 0.1) in *Dnmt1*, *Dnmt3a*, *Hdac2*, and *Tet2*. Moreover, the expression of these genes in preclinical mice was also significantly lower than the one observed in clinical mice, the difference being statistically significant for *Dnmt3b* (*p* < 0.05) and *Hdac1* (*p* < 0.05), and with a trend to signification for *Dnmt1* (*p* = 0.1). On the contrary, no significant differences were observed between clinical mice and their controls (Figure 2). Ct values of the RT-qPCR analysis are presented in Appendix A.

In order to test if the lack of statistical significance was due to age differences within the group that could modify gene expression, we calculated the correlation between gene expression Ct values and age in controls. No significant Pearson correlation values were found (*p* > 0.05) for any of the analyzed genes.

#### 2.2.2. Gene Expression Profile in Sheep

In sheep, the expression profile of six genes (*DNMT1*, *DNMT3A*, *DNMT3B*, *HDAC1*, *HDAC2*, and *TET1*) was also analyzed by RT-qPCR in the thalamus region. The sheep were divided into three groups of study: preclinical scrapie-infected sheep, clinical scrapie-infected sheep, and healthy control sheep.

No significant differences were observed in the expression levels of *DNMT1*, *DNMT3A*, and *DNMT3B* genes between the different groups of animals. However, a trend to signification (*p* = 0.1) between control and preclinical sheep, and between control and clinical sheep was observed in *TET1*, whose expression levels, just like in mice, were lower in clinical and preclinical sheep compared to the control ones. In addition, downregulation of *HDAC2* was observed in preclinical sheep compared to controls and clinical animals (*p* < 0.05). These expression trends in the *HDAC2* gene were like those observed in preclinical Tg338 mice. A downregulation of *HDAC1* expression was also observed in clinical scrapie sheep compared to controls (*p* = 0.1), in contrast with the increased expression displayed by the clinical Tg338 mice (Figure 3). Ct values of the RT-qPCR analysis are shown in Appendix A. Similar to mice, gene expression levels did not correlate with age in controls.

### 2.3. 5-Methylcytosine and 5-Hydroxymethylcytosine Brain Profiles of Tg338 Mice and Sheep

#### 2.3.1. 5mC and 5hmC Levels in Tg338 Mice Brains

Immunohistochemistry was performed in CNS tissue sections from Tg338 mice for the detection of 5mC and 5hmC. Nine brain areas were studied in the four groups of mice: the frontal cortex, parietal cortex, thalamus, hypothalamus, hippocampus, mesencephalon, cerebellum, obex, and septal area/striatum.

In all the studied brain areas, 5mC and 5hmC displayed intranuclear staining in neurons and glial cells.

Regarding 5mC levels, clinical and preclinical mice displayed similar levels across all the studied brain areas (Figure 4). The same trend was observed between preclinical mice and their respective controls (Figure 4). The obex was the only area that showed a significant difference, namely a decrease in 5mC levels in clinical compared to control mice (*p* < 0.05) (Figure 4 and Figure 5). To determine whether the lack of significance between the different groups was a consequence of differences in age at the sacrifice, we calculated Pearson’s correlation between age and immunohistochemical parameters in the animals of control groups, finding only a significant positive correlation (*p* < 0.05) in the cerebellum (r = 0.68) and hypothalamus (r = 0.70).

A different pattern was observed for 5hmC. Clinical mice showed similar levels to their controls in all studied brain areas (Figure 6). In contrast, 5hmC levels were increased in preclinical mice compared to their controls and the clinical mice (Figure 6). This increase was significant in some of the analyzed brain areas, including in the parietal cortex and cerebellum between preclinical and control mice (*p* < 0.05) and in the thalamus (*p* < 0.05), hypothalamus (*p* < 0.01), parietal cortex (*p* < 0.01), and cerebellum (*p* < 0.05) between preclinical and clinical mice (Figure 6 and Figure 7). Hydroxymethylation did not display a significant correlation with age in any of the analyzed areas (Pearson’s correlation *p* > 0.05).

#### 2.3.2. 5mC and 5hmC Levels in Scrapie Sheep Brains

In sheep CNS tissue sections, immunohistochemistry was performed to detect 5mC and 5hmC in eight areas: the frontal cortex, basal ganglia cortex, basal ganglia, parietal cortex, thalamus, hippocampus, mesencephalon, and obex.

Both 5mC and 5hmC showed nuclear and perinuclear staining in neurons and glial cells. The levels of 5mC negatively correlated with age in the thalamus of control sheep (r = −0.95, *p* = 0.02). No significant correlations were found in any other areas for 5mC or 5hmC.

Like what was observed in mice, clinical and preclinical sheep showed similar 5mC levels in all studied brain areas (Figure 8). However, infected animals displayed lower values of 5mC signal than controls (Figure 8). This reduction was significant in the mesencephalon (*p* < 0.05), thalamus (*p* < 0.01), and parietal cortex (*p* < 0.01) of clinical scrapie sheep (Figure 8 and Figure 9), and in the mesencephalon (*p* < 0.05), obex (*p* < 0.05), thalamus (*p* < 0.05), and parietal cortex (*p* < 0.01) of preclinical animals (Figure 8 and Figure 9).

Interestingly, 5hmC displayed the same pattern as 5mC. Contrary to the changes found in mice, no differences were observed in the levels of 5hmC between clinical and preclinical sheep (Figure 10), and lower levels of 5hmC were found in both clinical and preclinical sheep in comparison to healthy animals (Figure 10). This decrease in 5hmC levels was significant in the mesencephalon (*p* < 0.01), obex (*p* < 0.05), thalamus (*p* < 0.01), and parietal cortex (*p* < 0.01) of clinical animals (Figure 10 and Figure 11), and also in the mesencephalon (*p* < 0.001), obex (*p* < 0.01), thalamus (*p* < 0.01), and parietal cortex (*p* < 0.01) of preclinical animals (Figure 10 and Figure 11).

### 2.4. Correlation between 5mC and 5hmC Levels

Pearson’s correlation was used to evaluate the relationship between 5mC and 5hmC. Comparing all the analyzed brain areas, 5mC and 5hmC displayed a weak but significant positive correlation in Tg338 mice (r = 0.1432, *p* = 0.0459) and a stronger correlation in sheep (r = 0.6136, *p* = 1.023 × 10^−12^), as shown in Figure 12a,b, respectively.

### 2.5. Correlation between the Levels of 5mC and 5hmC and the Expression Levels of Epigenetic Regulatory Genes in Tg338 Mice and Sheep

In the thalamus region of Tg338 mice and sheep brains, some associations were observed between 5mC and 5hmC levels, and several genes were implicated in epigenetic regulation using Pearson’s correlation. Most of these associations were related to 5hmC levels.

In Tg338 mice, significant positive correlations were observed between 5hmC and *Dnmt1* (r = 0.5559, *p* = 0.0166) (Figure 13a), *Dnmt3a* (r = 0.6579, *p* = 0.0016) (Figure 13b), *Tet1* (r = 0.6479, *p* = 0.0027) (Figure 13c), *Tet2* (r = 0.5290, *p* = 0.0199) (Figure 13d), and *Hdac1* (r = 0.5861, *p* = 0.0066) (Figure 13e).

On the contrary, significant negative correlations between *HDAC2*, 5mC (r = −0.5538, *p* = 0.0322) (Figure 13f), and 5hmC levels (r = −0.6278, *p* = 0.0122) (Figure 13g) were observed in sheep.

### 2.6. Correlation of 5mC and 5hmC Levels with Prion-Related Lesions

Spearman’s correlation revealed some associations between 5mC, 5hmC, and prion-related lesions when comparing all the analyzed brain areas in the full set of animals, especially in sheep.

Vacuolation was negatively correlated with 5mC in Tg338 mice (ρ = −0.1528, *p* = 0.0295). In sheep, 5mC displayed a negative correlation with PrP^Sc^ deposition (ρ = −0.3082, *p* = 0.0015) and vacuolation (ρ = −0.4369, *p* = 0.000002). 5hmC was also negatively correlated with vacuolation (ρ = −0.4167, *p* = 0.000006) and PrP^Sc^ accumulation (ρ = −0.3731, *p* = 0.00006). However, these significant correlations were lost when this parameter was calculated using only the set of scrapie-infected animals.

## 3. Discussion

Although recent evidence suggests an involvement of DNA methylation in prion disease pathology, the knowledge about its specific functions and roles is still limited. In the current study, we assessed the 5mC and 5hmC immunohistochemical brain profiles and the expression levels of genes encoding DNA methylation enzymes in a murine model of scrapie disease (Tg338 mice) and sheep naturally infected with scrapie. Moreover, we analyzed possible associations between the 5mC and 5hmC levels with prion-related lesions and with the expression levels of the DNA methylation-related genes.

The levels of 5mC have been studied in other neurodegenerative diseases, such as AD. In the brain of a triple-transgenic mouse model of AD (3xTg-AD) [37] and the brain of preclinical AD patients [38], a decrease in this epigenetic modification was reported. Similar results have been observed in our study, where scrapie-infected sheep and mice showed similar patterns. Clinical and preclinical animals displayed comparable brain levels of 5mC, which were significantly lower in the obex of clinical mice and the mesencephalon, thalamus, parietal cortex, and obex of clinical and preclinical sheep. Furthermore, a negative association of 5mC levels with prion-related lesions was observed in mice and especially in sheep, in which a negative correlation of 5mC with PrP^Sc^ accumulation and vacuolation was found, although the statistical significance of this correlation was lost when considering only the set of scrapie animals. This decrease in 5mC immunostaining is not related with the loss of cells due to prion toxicity and the consequent loss of nuclei because immunoreactivity was normalized taking into account the number of nuclei in each area. As a result, the decrease must be due to a real decrease in immunostaining inside the nucleus of the different cell populations. Although experimental animals used here displayed a similar age, small age differences at animal sacrifice could possibly have an effect on DNA methylation. Within the control group, age variability was found significantly correlated with 5mC in the thalamus of the control sheep. Nevertheless, the reduction in 5mC was still significant in the scrapie thalamus. In control mice, 5mC correlated with age in hypothalamus and cerebellum. We cannot discard the possibility that significant differences in these two specific areas of CNS between control and scrapie mice could have been observed in more homogenous groups. If the decline observed in 5mC levels is a trigger of the disease or a consequence of the disease-related neurodegenerative lesions, it is still unclear and warrants further investigation.

Several reports have shown distinctive levels of 5hmC in Alzheimer’s and Parkinson’s diseases with different trends between studies. In preclinical and clinical AD patients, different brain regions have been shown to display higher levels of 5hmC [20,38,39]. This increment was also observed in 3xTg-AD [37] and amyloid precursor protein (APP)/presenilin 1 (PS1) [40] AD mouse models and in the cerebellum of PD patients [22,41]. However, other studies point to a decrease in 5hmC in the entorhinal cortex and cerebellum of AD patients [42] and the brain of 3xTg-AD [43] and APP/PS1 [44] mice. In the present study, contrarily to 5mC, 5hmC levels displayed opposite results between sheep and mice. An increase in 5hmC levels was found in preclinical mice, whereas in clinical and preclinical sheep, 5hmC levels decreased in different CNS areas. Differences between the experimental and the natural model may be due to differences in the stage of disease in preclinical animals, whereas a controlled scrapie inoculation in mice allows sacrificing animals in a true preclinical stage clearly distinct from the clinical phase; preclinical sheep can be detected in a late stage, closer to the onset of symptoms.

Although no association was found between prion lesions and 5hmC in mice, in sheep, both 5mC and 5hmC correlated negatively with PrP^Sc^ deposits and vacuolation. Similarly, these correlations were not observed in the set of scrapie animals, suggesting that the correlation is linked to the course of the disease but not to the degree of the lesion. In the natural model, 5hmC follows the same decrease trend as 5mC. This positive correlation between 5mC and 5hmC has been reported in other studies of patients with autism [45] and AD [20]. As opposed to a negative correlation that could be related to the increment of 5hmC as a consequence of the decline in 5mC levels, the concurrent decrease in 5mC and 5hmC suggests that 5hmC could have a specific role in prion disease pathology and not only act as a demethylation intermediate of 5mC.

Differential expression of genes encoding epigenetic regulatory enzymes was also observed. *Dnmt3b* interacts with *Hdac1* during the establishment of DNA methylation patterns [46]. Moreover, along with its role in de novo DNA methylation, *Dnmt3b* also functions as a DNA dehydroxymethylase, able to directly convert 5hmC to an unmethylated cytosine (C) [47]. In our study, preclinical mice displayed downregulation of *Dnmt3b* whereas, in clinical mice, *Dnmt3b* expression returns to normal levels. This early downregulation was not associated with a decrease in 5mC in the thalamus. In sheep, neither *DNMT3B* expression nor the levels of 5mC were modified significantly between preclinical and clinical animals. These expression changes are neither related to an increase in 5hmC, on the contrary, a significant decrease in 5hmC was observed in both, mice and sheep infected with scrapie. Modification in gene expression could be related with other biological processes. In the adult brain, *Dnmt3b* is required for neurogenesis, facilitating the neuronal maturation in the hippocampus of adult mice [48], and it also plays a role in regulating object-place recognition memory [49]. *Hdac1* is needed as well for neuronal differentiation of murine hippocampal neural stem cells [50] and is involved in DNA damage repair pathways and cognitive function [51], including fear extinction learning [52]. This latter brain function is compromised in schizophrenia patients [52]. Upregulation of *HDAC1* has been found in the prefrontal cortex of patients with schizophrenia [53] and overexpression of this enzyme seems to ameliorate the fear extinction learning cognitive function in mice [52]. The upregulation of *Hdac1* observed in clinical mice could be a compensatory mechanism trying to maintain balanced levels of 5mC and 5hmC, and at the same time, counteract the neurodegenerative effects produced by scrapie disease.

A recent study in mouse embryos showed that *Hdac1* and *Hdac2* were essential for maintaining correct DNA methylation patterns during preimplantation development. Deficiency of *Hdac1* and *Hdac2* in these embryos caused an increase in both forms of DNA methylation, 5mC and 5hmC [54]. Therefore, the upregulation of *HDAC2* observed in clinical scrapie sheep could contribute to the decrease in 5mC and 5hmC levels. Upregulation of *HDAC2* has also been observed in the brain of patients with AD [55] and PD [56] and Swiss albino old mice, and *HDAC2* overexpression has been correlated with reduced recognition memory [57,58]. Inhibition of *HDAC2*, conversely, slows down AD progression through ameliorating amyloid beta-induced neuronal impairments in APP/PS1 mice [59], enhancing mitochondrial respiration and reducing the levels of neurotoxic amyloid beta peptides in induced pluripotent stem cell-derived neurons [60]. These findings suggest that upregulation of *HDAC2*, in addition to contributing to the decline of 5mC and 5hmC, may also be entailed in aggravating the neurodegenerative process. Further studies are essential for delving into the specific role of this enzyme in prion diseases and studying its potential as a therapeutic target.

On the other hand, *Tet1*, which was downregulated in preclinical mice and preclinical and clinical sheep, is one of the enzymes capable of the conversion of 5mC to 5hmC, as well as its further oxidation to 5fC and 5caC intermediates [61]. The relationship between *Tet1* expression and 5hmC levels is somewhat controversial. Decreased levels of *TET1* accompanied by increased levels of 5hmC have been observed in chondrocytes from patients suffering osteoarthritis due to the minor conversion of 5hmC to 5fC or 5caC [62]. Cancer-related studies, however, show an association between reduced *TET1* expression levels and a decrease in 5hmC [63,64]. In addition, this enzyme has important functions in the adult brain beyond its role in DNA demethylation, being necessary for a functional axon-myelinic interface and a successful myelin repair [65] and also critical in the regulation of the neuroinflammation process, associating a downregulation of *Tet1* with aberrant activation of inflammatory response pathways [66]. Interestingly, neuroinflammation is increased during the course of scrapie disease [67,68,69,70]. In addition, *Tet1* knock-out mice show a deficiency in adult neurogenesis, spatial learning, and memory [71]. The downregulation of *Tet1* during scrapie disease could have different functional implications regarding the regulation of 5hmC levels, which are positively correlated with the levels of *Tet1* in mice and decrease when the expression of this gene is downregulated and could also be involved in the impairment of the myelin repair and neuroinflammation pathways and the disruption of the neurogenesis process, contributing altogether to the disease progression.

In conclusion, our results show that DNA methylation and its regulating enzymes are involved in scrapie disease pathology, although there are differences in the epigenetic regulation between the natural model of the disease and the transgenic model. Both methylation forms, 5mC and especially 5hmC, were altered across different brain regions in the two models. 5mC diminished in infected mice and sheep and opposite trends were found for 5hmC, which increased in preclinical mice but diminished in preclinical and clinical sheep. The correlation found between 5mC and 5hmC suggests that the latter could have a relevant role in prion pathogenesis, not only as the intermediate derivative of 5mC. Differential expression of genes encoding epigenetic enzymes was also observed and related mainly to 5hmC patterns. Further studies are warranted to uncover the roles of 5mC and especially 5hmC in prion pathology. Oxidative bisulfite sequencing approaches would be necessary in order to quantify exactly the brain levels of 5hmC and 5mC and to identify specific genes with differential methylation or hydroxymethylation. More research is also indispensable to unravel the exact functions of the different enzymes of the epigenetic machinery in the disease progression and their potential for therapeutic interventions.

## 4. Materials and Methods

### 4.1. Animals

#### 4.1.1. Tg338 mice

Tg338 mice (*n* = 24), which express a transgenic VRQ allele of the ovine *PRNP* gene under the ovine PrP promoter [35,36], were used in this study. The mice were divided into four groups: clinical mice (*n* = 6), preclinical mice (*n* = 6), clinical control (*n* = 6), and preclinical control (*n* = 6). Clinical and preclinical mice were intracerebrally inoculated with Tg338-adapted classical scrapie CNS homogenate, originally derived from ARQ/ARQ scrapie sheep, while preclinical and clinical controls were mock inoculated with non-inoculated Tg338 CNS homogenate, using the procedure described earlier [72].

The animals included in the preclinical group were sacrificed before showing symptoms associated with scrapie and the clinical ones once they already showed these symptoms. At the same time, the control animals of each group were sacrificed at similar times. Animals in the preclinical groups were inoculated approximately 6 weeks later than the clinical groups in order to sacrifice all animals at similar ages and avoid a possible effect of age in the observed epigenetic modifications [73,74]. In an attempt to achieve homogeneous groups, controls and preclinical mice were sacrificed at different times corresponding to the days on which the animals in the clinical group were sacrificed, when each of them showed clear symptoms of prion disease. The characteristics of the Tg338 mice are summarized in Table 1.

All experimental procedures were carried out in compliance with the recommendations for the care and use of experimental animals established by Spanish law (R.D. 53/2013) and European Directive 2010/63/UE, and were approved by the Ethics Committee for Animal Experimentation of the University of Zaragoza (PI40/15, PI138/15).

#### 4.1.2. Sheep

A total of 15 female Rasa Aragonesa sheep, aged between 4 and 6 years and carrying the ARQ/ARQ genotype for the ovine *PRNP* gene, were included in this study. They were divided into three groups depending on their clinical status: clinical sheep (*n* = 5), preclinical sheep (*n* = 5), and control sheep (*n* = 5). All selected animals had similar ages in order to exclude a possible influence of age in the observed epigenetic modifications [73,74]. Sheep in the preclinical stage were identified by immunohistochemical analysis of rectal mucosa biopsies and sacrificed before clinical signs were detectable by pentobarbital overdose, whereas animals in the clinical stage were identified by the observation of scrapie-related symptoms. These animals correspond to those used previously in an assessment study of neurogranin and neurofilament light chain as preclinical biomarkers in scrapie [34].

### 4.2. Tissue Collection

After the euthanasia of Tg338 mice, the brain of each mouse was harvested and divided sagittally. One hemisphere was fixed by immersion in 10% formalin for up to 48 h for further histopathological and immunohistochemical analyses. The other hemisphere was frozen immediately in dry ice and conserved at −80 °C. From this hemisphere, the thalamus region was used for gene expression studies.

Regarding ovine samples, after sheep were sacrificed, samples from eight areas of the CNS (the frontal cortex, basal ganglia cortex, basal ganglia, parietal cortex, thalamus, hippocampus, mesencephalon, and obex) were collected and fixed in 10% formaldehyde for further use in histopathological and immunohistochemical analyses. In addition, thalamus samples from each sheep were preserved in RNAlater^TM^ solution (Thermo Fisher Scientific, Waltham, MA, USA) for gene expression analyses.

### 4.3. Neuropathological Evaluation of Tg338 Mice and Scrapie Sheep

Tg338 brains fixed previously in formalin were embedded in paraffin wax, cut into 4-µm-thick sections, and mounted on glass slides for evaluation of vacuolation, a prion-related lesion, using hematoxylin and eosin (H-E) staining. PrP^Sc^ deposition analysis was carried out using the paraffin-embedded tissue (PET) blot method previously described [75]. Briefly, paraffin-embedded sections were collected onto nitrocellulose membranes (0.45 μm pore size; Bio-Rad, Hercules, CA, USA) and digested with 250 μg/mL of proteinase K for 2 h at 56 °C. Proteins adhered to the membrane were denatured with 3 M guanidine thiocyanate (Sigma-Aldrich, St. Louis, MO, USA) and PrP^Sc^ was detected using Sha31 mouse monoclonal antibody (SPI-Bio, Montigny le Bretonneux, France) at 1:8000 dilution for 1 h. Sections were then incubated for 1 h with an alkaline phosphatase-coupled goat anti-mouse antibody (Dako, St. Clara, CA, USA) at 1:500 dilution. Enzymatic activity was visualized using NBT/BCIP chromogen (Sigma-Aldrich, St. Louis, MO, USA). Vacuolation and PrP^Sc^ deposition were semiquantitatively evaluated per area with a scale score, being 0 for lack of vacuoles/PrP^Sc^ deposit and 5 for very intense vacuolation/PrP^Sc^ deposit, following the standard method to assess these features [76]. The histopathological data of PrP^Sc^ deposition and vacuolation are shown in Figure 1 and Appendix A. In scrapie sheep, paraffin-embedded 4 µm-thick sections from the eight previously mentioned areas were used for histopathological evaluation of vacuolation by H-E staining. PrP^Sc^ was detected by immunohistochemistry using the monoclonal primary antibody L42 (R-Biopharm, Darmstadt, Germany) at 1:500 dilution for 30 min after formic acid treatment and proteinase K digestion, as previously described [77]. The histopathological evaluation of these sheep was performed in a previous study [34].

### 4.4. Expression Analysis of Genes Involved in Epigenetic Regulation

#### 4.4.1. Gene Expression Analysis in Tg338 Mice

Seven genes, described to be involved in epigenetic regulation, were selected for the analysis of their expression profile in the thalamus of Tg338 mice: *Dnmt1*, *Dnmt3a*, *Dnmt3b*, *Hdac1*, *Hdac2*, *Tet1*, and *Tet2*. Table 2 shows TaqMan assays (Thermo Fisher Scientific) used for the amplification of the genes of interest.

Total RNA was isolated using the RNeasy Lipid Tissue Mini kit (Qiagen, Valencia, CA, USA). The quality and quantity of RNA were determined using a NanoDrop instrument (Thermo Fisher Scientific, Waltham, MA, USA). An amount of 200 ng of total RNA was used for retrotranscription with qScript^TM^ cDNA Supermix (Quanta Biosciences^TM^, Gaithersburg, MD, USA), according to the manufacturer’s instructions. The resulting complementary DNA (cDNA) was diluted 1:5 in water and gene expression was quantified by RT-qPCR using the TaqMan universal PCR master mix assays (Thermo Fisher Scientific) in a StepOne Plus Real-Time PCR instrument (Applied Biosystems, Waltham, MA, USA). All reactions were run in triplicate. The comparative quantification of the results was standardized by the 2^−∆∆Ct^ method [78], using the geometric mean of *Sdha* and *H6pd* housekeeping genes (Table 2) as a normalizer. Student’s *t*-test was applied to identify differences between groups, which were considered significant at *p* < 0.05.

#### 4.4.2. Gene Expression Analysis in Sheep

The expression profile of six genes (*DNMT1*, *DNMT3A*, *DNMT3B*, *HDAC1*, *HDAC2*, and *TET1*) was assessed in sheep thalamus samples. The primers used for the RT-qPCR assay are listed in Table 3.

Total RNA isolation and retrotranscription were performed following identical procedures to the ones used with murine samples. The resulting cDNA was diluted 1:5 in water and gene expression was quantified by RT-qPCR using the Fast SYBR^TM^ Green Master Mix (Applied Biosystems, Thermo Fisher Scientific) in a StepOne Plus Real-Time PCR instrument (Applied Biosystems). Each PCR was performed in triplicate. The results were standardized by the 2^−∆∆Ct^ method [78], using the geometric mean of *SDHA* and *G6PDH* housekeeping genes [79] as a normalizer. To identify differences between groups, Student’s *t*-test was performed, and significant differences were considered at *p* < 0.05.

### 4.5. Immunohistochemical Analysis of 5-Methylcytosine and 5-Hydroxymethylcytosine Levels in Tg338 Mice and Sheep

Immunohistochemistry for the detection of 5mC and 5hmC was performed in paraffin-embedded CNS tissue sections from Tg338 mice and sheep. In Tg338 mice, nine brain areas were studied: the frontal cortex, parietal cortex, thalamus, hypothalamus, hippocampus, mesencephalon, cerebellum, obex, and septal area/striatum. In sheep, a total of eight brain areas were analyzed: the frontal cortex, basal ganglia cortex, basal ganglia, parietal cortex, thalamus, hippocampus, mesencephalon, and obex.

After deparaffination and rehydration, tissue sections were subjected to antigen retrieval with citrate buffer (pH 6.0) for 20 min at 85 °C in a PTLink (Dako). Afterward, endogenous peroxidase activity was blocked using a precast solution (Dako Agilent, Glostrup, Denmark). Sections were then incubated overnight at 4 °C with the primary antibodies 5mC (Thermo Fisher Scientific, AB_2787107) at 1:100 dilution in Tg338 mice and 1:75 in sheep and 5hmC (Thermo Fisher Scientific, AB_2610634) at 1:200 dilution in Tg338 mice and 1:75 in sheep. The omission of the primary antibody served as a background control for the nonspecific binding of the secondary antibody. Subsequently, the sections were incubated with an anti-mouse enzyme-conjugated EnVision polymer (Dako Agilent, Glostrup, Denmark) for 30 min at room temperature. Diaminobenzidine (DAB, Dako Agilent, Glostrup, Denmark) was used as the chromogen.

CNS sections were assessed and photographed using a Zeiss Axioskop 40 optical microscope (Zeiss, Jena, Germany). 5mC and 5hmC immunostaining levels were evaluated using Image J software and following the semi-quantification analysis method [80], in which for each image, the DAB staining intensity is normalized by the number of nuclei. Afterward, the normalized DAB staining intensity mean and its standard deviation were calculated for each group of animals (clinical, preclinical, and control) in both Tg338 mice and sheep. Student’s *t*-test was applied to identify differences between scrapie-infected and control groups, which were considered significant at *p* < 0.05.

### 4.6. Correlation between 5mC and 5hmC Levels, Age, Gene Expression, and Prion-Associated Features

Pearson’s and Spearman’s correlations calculated with GraphPad Prism software v.5 were used to evaluate possible relationships between 5mC and 5hmC intensities and the expression levels of epigenetic regulatory genes in the thalamic region, and between 5mC and 5hmC levels and prion-related lesions in both Tg338 mice and sheep. Correlations were considered significant at *p* < 0.05. Correlations between the different methylation parameters and age were also calculated in control groups of mice and sheep in order to evaluate if age differences within groups could affect the obtained results.

## Figures and Tables

**Figure 1 ijms-24-01621-f001:**
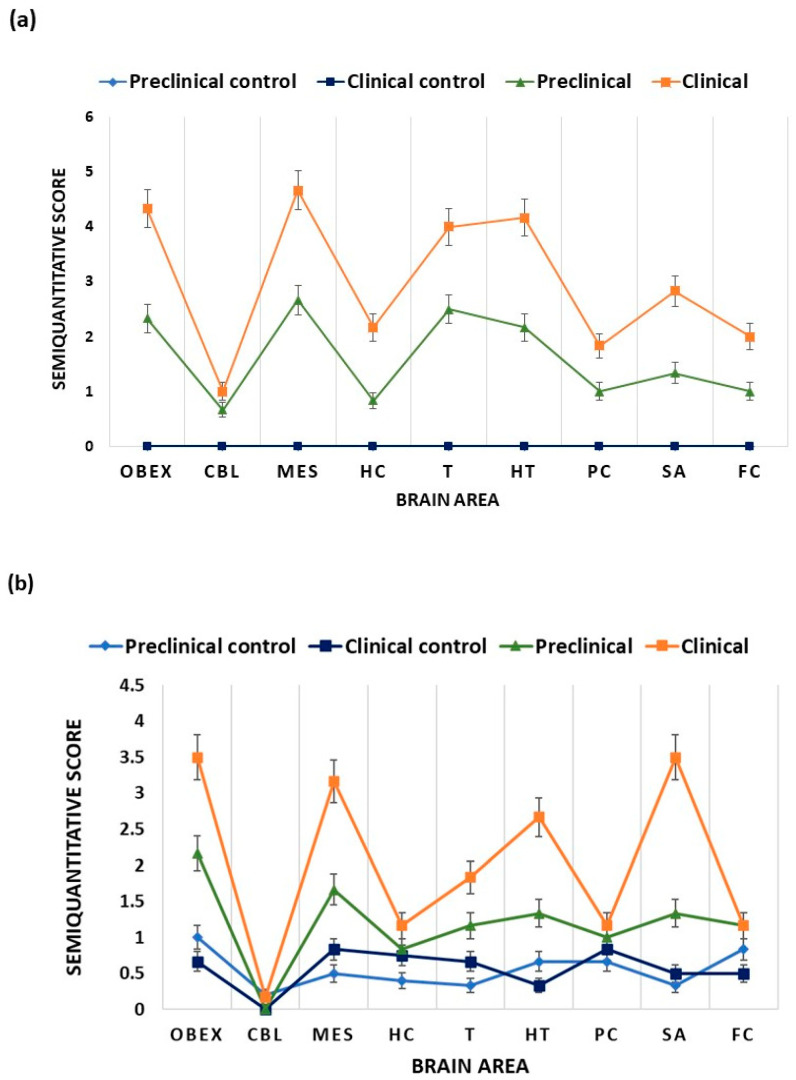
Semiquantitative evaluation in Tg338 mice of (**a**) PrP^Sc^ deposits and (**b**) vacuolation on a semiquantitative scale being 0 for lack of staining/vacuolation and 5 for the staining/vacuolation present at maximum intensity. In (**a**) preclinical control and clinical control display the same levels of PrP^Sc^ deposits. Data are shown as mean values ± SEM in the following brain areas: OBEX, cerebellum (CBL), mesencephalon (MES), hippocampus (HC), thalamus (T), hypothalamus (HT), parietal cortex (PC), septal area/striatum (SA), and frontal cortex (FC).

**Figure 2 ijms-24-01621-f002:**
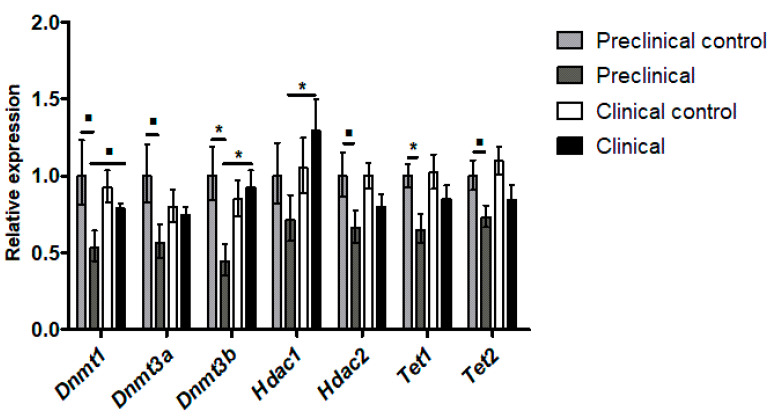
Relative expression levels in terms of 2^−∆∆Ct^ of epigenetic regulatory genes in the thalamus of Tg338 mice. ▪ *p* = 0.1; * *p* < 0.05.

**Figure 3 ijms-24-01621-f003:**
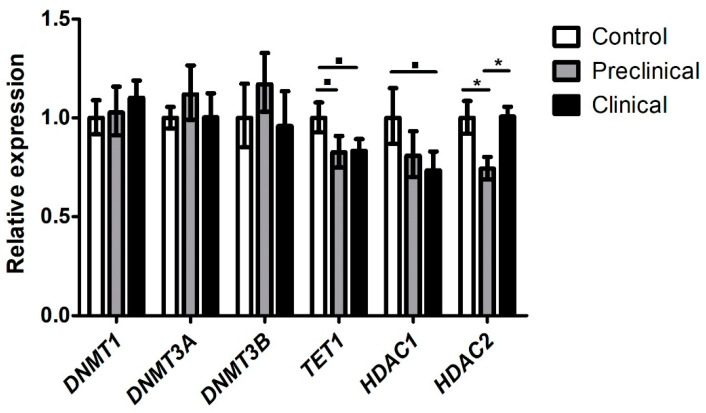
Relative expression levels in terms of 2 ^−∆∆Ct^ of genes involved in epigenetic regulation in the thalamus of sheep. ▪ *p* = 0.1; * *p* < 0.05.

**Figure 4 ijms-24-01621-f004:**
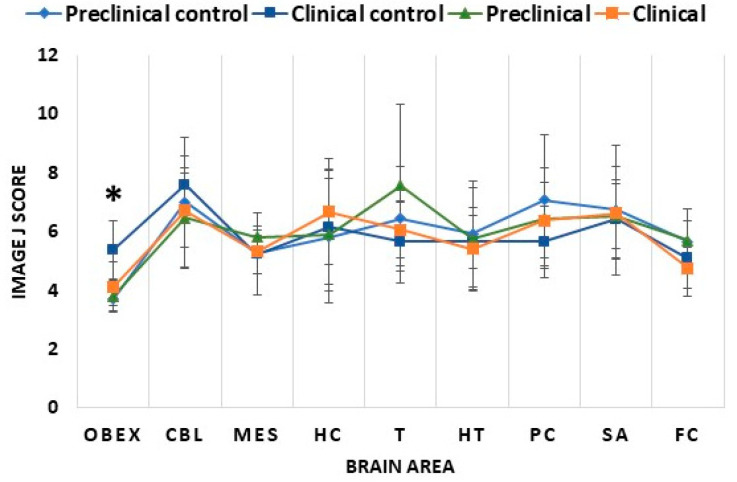
Comparison of 5mC Image J scores of the studied brain areas in the different mice groups: preclinical control, clinical control, preclinical, and clinical mice. The nine studied areas were the obex, cerebellum (CBL), mesencephalon (MES), hippocampus (HC), thalamus (T), hypothalamus (HT), parietal cortex (PC), septal area/striatum (SA), and frontal cortex (FC). The data are presented as mean values ± SEM. * Significant difference between clinical control and clinical mice (*p* < 0.05).

**Figure 5 ijms-24-01621-f005:**
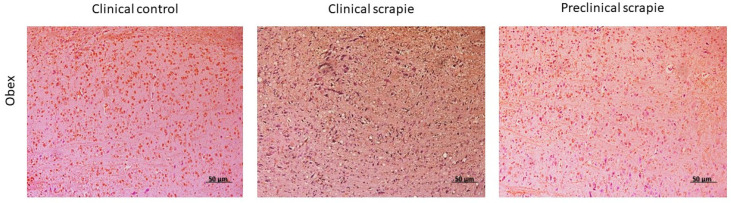
Representative images of 5mC immunostaining in the obex of clinical control mice and clinical and preclinical mice.

**Figure 6 ijms-24-01621-f006:**
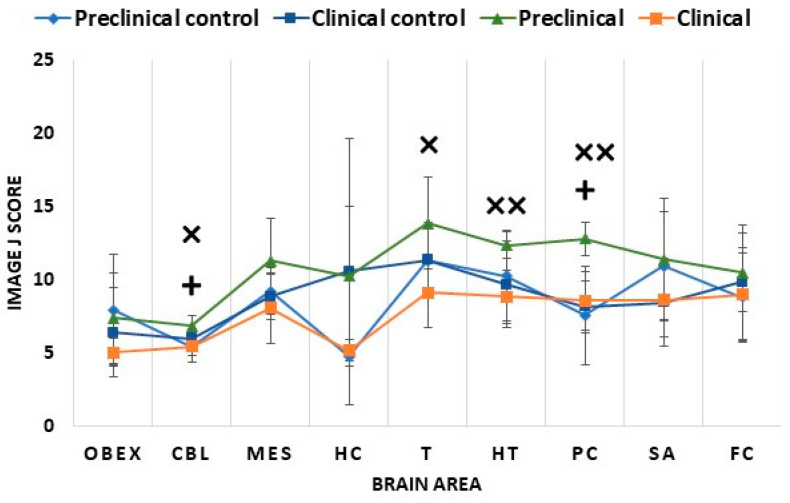
Comparison of 5hmC Image J scores of the studied brain areas in the different mice groups: preclinical control, clinical control, preclinical, and clinical mice. The nine studied areas were the obex, cerebellum (CBL), mesencephalon (MES), hippocampus (HC), thalamus (T), hypothalamus (HT), parietal cortex (PC), septal area/striatum (SA), and frontal cortex (FC). The data are presented as mean values ± SEM. + Significant difference between preclinical control and preclinical mice (*p* < 0.05); × significant difference between preclinical and clinical mice (*p* < 0.05); ×× significant difference between preclinical and clinical mice (*p* < 0.01).

**Figure 7 ijms-24-01621-f007:**
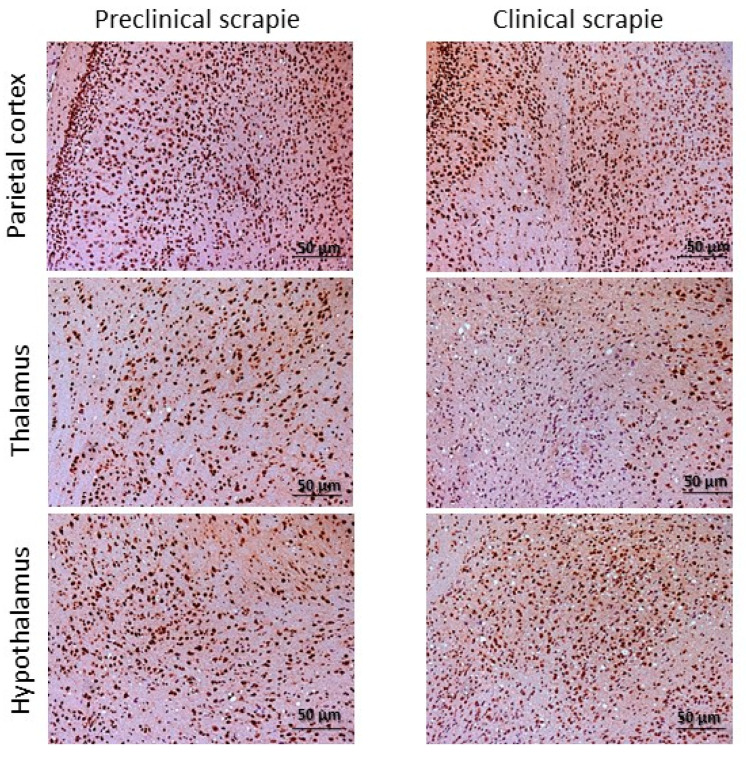
Immunostaining patterns of 5hmC in the parietal cortex, thalamus, and hypothalamus of preclinical and clinical Tg338 mice.

**Figure 8 ijms-24-01621-f008:**
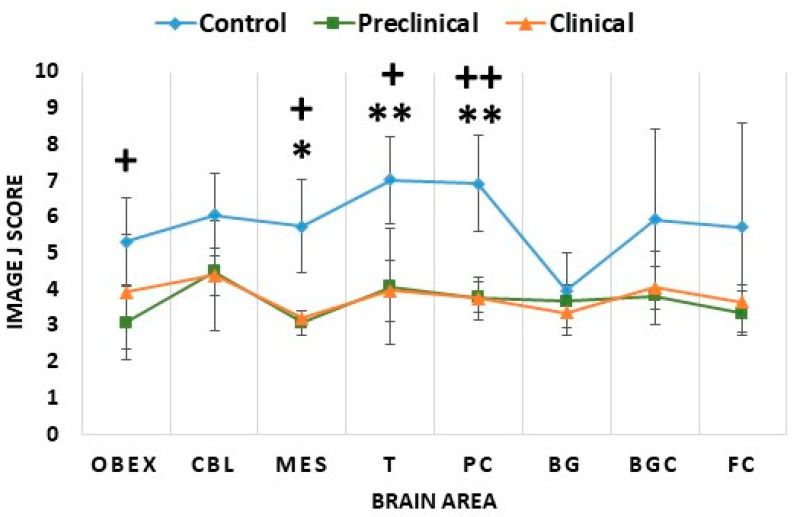
Comparison of 5mC Image J scores in the three sheep groups: control, preclinical, and clinical. The analyzed brain areas were the obex, cerebellum (CBL), mesencephalon (MES), thalamus (T), parietal cortex (PC), basal ganglia (BG), basal ganglia cortex (BGC), and frontal cortex (FC). The data are presented as mean values ± SEM. * Significant difference between control and clinical sheep (*p* < 0.05); ** significant difference between control and clinical sheep (*p* < 0.01); + significant difference between control and preclinical sheep (*p* < 0.05); ++ significant difference between control and preclinical sheep (*p* < 0.01).

**Figure 9 ijms-24-01621-f009:**
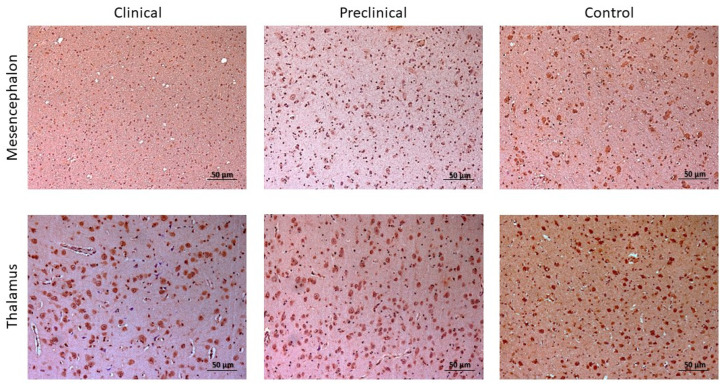
Immunostaining determination of 5mC in mesencephalon and thalamus of preclinical, clinical, and control sheep.

**Figure 10 ijms-24-01621-f010:**
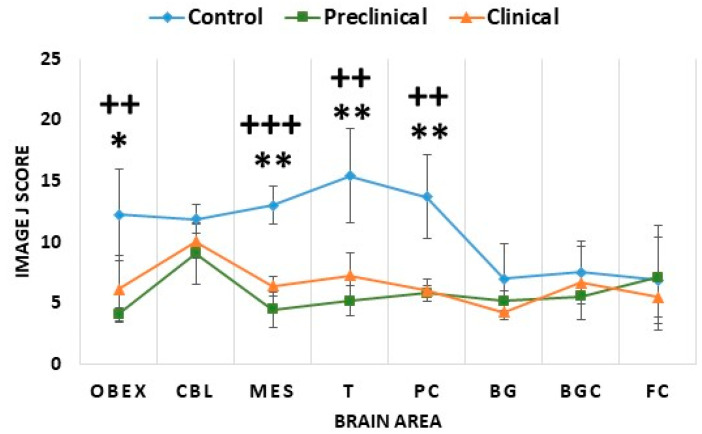
Comparison of 5hmC Image J scores in the three sheep groups: control, preclinical, and clinical. The analyzed brain areas were the obex, cerebellum (CBL), mesencephalon (MES), thalamus (T), parietal cortex (PC), basal ganglia (BG), basal ganglia cortex (BGC), and frontal cortex (FC). The data are presented as mean values ± SEM. * Significant difference between control and clinical sheep (*p* < 0.05); ** significant difference between control and clinical sheep (*p* < 0.01); ++ significant difference between control and preclinical sheep (*p* < 0.01); +++ significant difference between control and preclinical sheep (*p* < 0.001).

**Figure 11 ijms-24-01621-f011:**
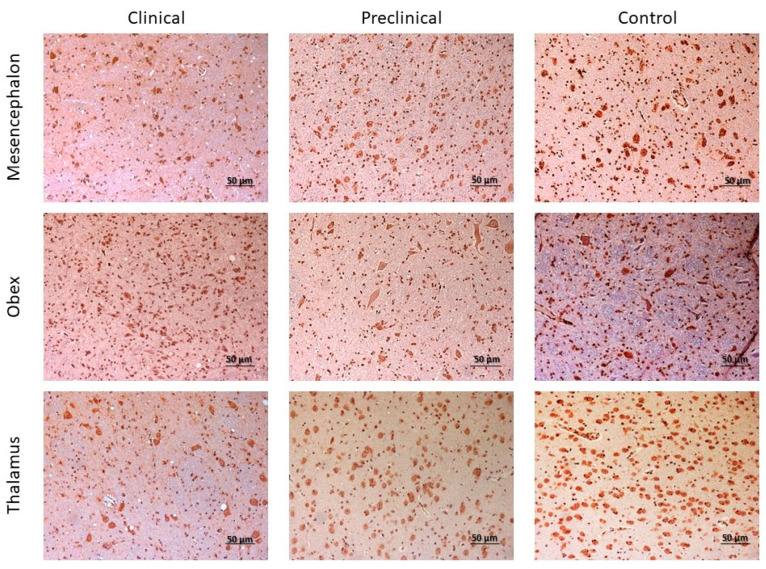
Immunostaining patterns of 5hmC in the mesencephalon, obex, and thalamus of preclinical, clinical, and control sheep.

**Figure 12 ijms-24-01621-f012:**
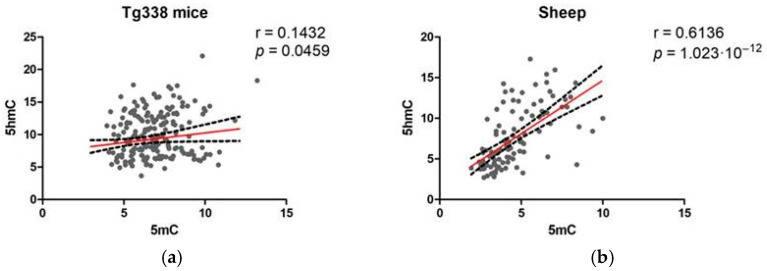
Pearson’s correlation between 5mC and 5hmC in (**a**) Tg338 mice and (**b**) sheep.

**Figure 13 ijms-24-01621-f013:**
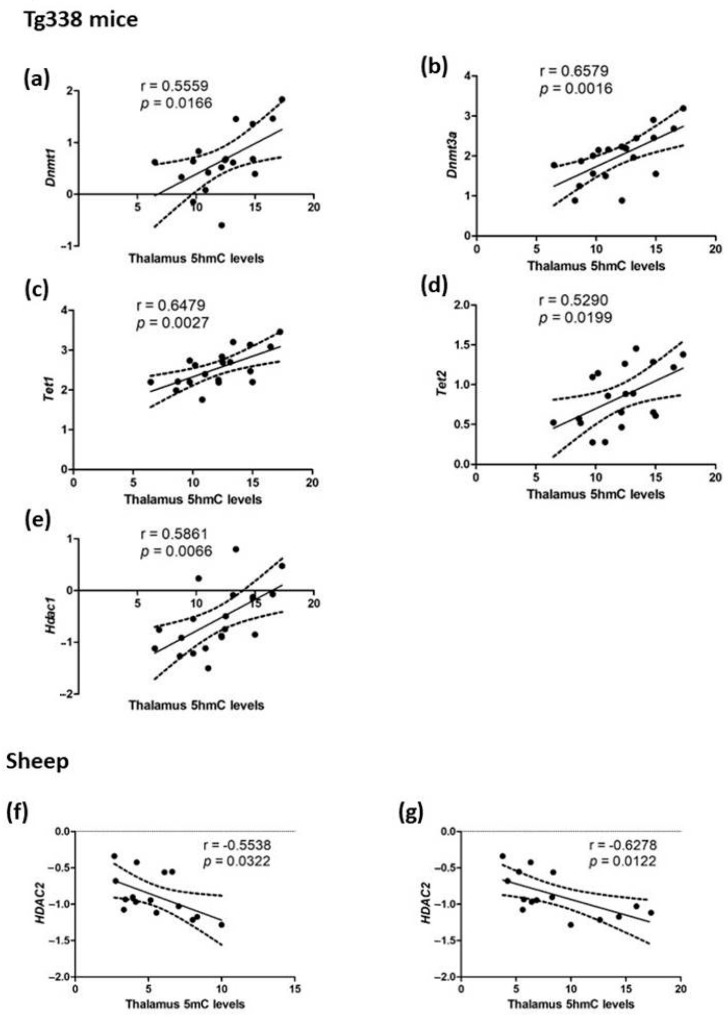
Correlations in the thalamus region of 5mC and 5hmC with the expression levels of some genes involved in epigenetic regulation. 5hmC positively correlated with (**a**) *Dnmt1*, (**b**) *Dnmt3a*, (**c**) *Tet1*, (**d**) *Tet2*, and (**e**) *Hdac1* in Tg338 mice, and 5mC (**f**) and 5hmC (**g**) negatively correlated with *HDAC2* in sheep.

**Table 1 ijms-24-01621-t001:** Characteristics of the Tg338 mice whose samples were used in this study. ID = mouse identification code and DPI = days post-inoculation at which they were sacrificed.

Group	ID	Gender	Age of Inoculation (Days)	Age of Sacrifice (Days)	DPI
Clinical control	7854-O	Male	42	216	174
	7850-O	Male	42	196	154
	6797-O	Male	42	190	148
	8756-O	Male	42	210	168
	8916-O	Male	42	212	170
	8258-O	Male	35	214	179
Clinical mice	7741-O	Male	35	189	154
	8562-O	Male	35	203	168
	7840-O	Male	35	209	174
	5720-O	Male	35	214	179
	0445-O	Male	35	205	170
	0527-O	Male	35	183	148
Preclinical control	6404-O	Male	77	214	137
	8373-O	Male	84	219	135
	8508-O	Male	84	196	112
	5997-O	Male	84	212	128
	6768-O	Male	84	212	128
	7038-O	Male	84	196	112
Preclinical mice	7731-O	Male	79	191	112
	6775-O	Male	79	214	135
	7000-O	Male	79	216	137
	7457-O	Male	79	207	128
	6248-O	Male	79	207	128
	6500-O	Male	79	191	112

**Table 2 ijms-24-01621-t002:** References of the probes used for the RT-qPCR TaqMan amplification assay in Tg338 mice. ID = identification of the commercial company (Thermo Fisher Scientific, Waltham, MA, USA).

Gene	ID
*Dnmt1*	Mm01151063_m1
*Dnmt3a*	Mm00432881_m1
*Dnmt3b*	Mm01240113_m1
*Tet1*	Mm01169087_m1
*Tet2*	Mm00524395_m1
*Hdac1*	Mm02745760_g1
*Hdac2*	Mm00515108_m1
*Sdha*	Mm01352366_m1
*H6pd*	Mm00557617_m1

**Table 3 ijms-24-01621-t003:** Sequences of primers used in the RT-qPCR assay in sheep. Fw = Forward primer and Rv = Reverse primer.

Gene	Primer Sequences
*DNMT1*	Fw: 5′-CCCAGGAGAAGCAAGTCTGATG-3′ Rv: 5′-TGATGGTGGTCTGCCTGGTAGT-3′
*DNMT3A*	Fw: 5′-GGGACCCCTACTACATCAGCAA-3′ Rv: 5′-GCATTCATTACTGCGATCACCTT-3′
*DNMT3B*	Fw: 5′-TGGTTTGGTGATGGCAAGTTC-3′ Rv: 5′-TGAAGGTCGCCAGGTTAAAGTG-3′
*HDAC1*	Fw: 5′-CCTCTCCGAGATGGGATTGA-3′ Rv: 5′-CTGCACTGGGCTGGAACAT-3′’
*HDAC2*	Fw: 5′-GGAGCCCATGGCGTACAGT-3′ Rv: 5′-ACCCTGTCCGTAGTAATAATTTCCA-3′
*TET1*	Fw: 5′GAAATGCAATAAGGATAGAAATAGTAGTGTACA-3′ Rv: 5′-TCTTCGTCGCTGCTTCTTCTT-3′

## Data Availability

The data presented in this study are available within the article text, figures, and Appendix A.

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
