# Peer review of "5-Methylcytosine and 5-Hydroxymethylcytosine in Scrapie-Infected Sheep and Mouse Brain Tissues"

_ijms, 2023, doi:10.3390/ijms24021621_

Round 1

Reviewer 1 Report

Hernaiz and colleagues evaluated the expression of key epigenetic regulatory enzymes at two stages of prion disease progression; preclinically and clinically scrapie sick.

The paper is sound and well written.

I wonder whether the lack of significance for some of the results presented is due to age: indeed, culling was not homogeneous and there is quite a wide variability of age  within each group and between groups.

The authors may wish to comment on the age.

I also would like to see the scrapie characterization as a main figure. 

Finally, when presenting relative expression for qPCR, one lose the exact level of expression. it would be good to see the CT values of the experiments presented in suppl material.

Author Response

We would like to thank this reviewer for the effort made to read and review this manuscript and for his/her comments that have contributed to improve it. You can find hereafter how we have addressed your suggestions. Reviewer comments are in bold and italic characters.

Hernaiz and colleagues evaluated the expression of key epigenetic regulatory enzymes at two stages of prion disease progression: preclinically and clinically scrapie sick.

The paper is sound and well written.

We would like to thank reviewer #1 for his/her favourable comments.

  1. I wonder whether the lack of significance for some of the results presented is due to age: indeed, culling was not homogeneous and there is quite a wide variability of age within each group and between groups. The authors may wish to comment on the age.

We would like to clarify how we have taken into account the age of animals in research design. All groups of animals were sacrificed at similar ages to avoid a possible effect of age on the observed epigenetic changes. However, trying to get homogeneous groups, controls and preclinical mice were sacrificed at different times corresponding to the days where every mice of the clinical group displayed clear symptoms of the prion disease. With this experimental design there are not significant differences between groups, however there is certain variability of age within groups.

To test if this variability could mask differences between groups, we calculated the Pearson and Spearman correlation between age and the different methylation parameters in mice and sheep, respectively. We have found statistically significant correlations between age and 5mC in specific regions of the CNS of sheep and mice. In ovine thalamus, the correlation found did not mask the differences observed between groups and in hypothalamus and cerebellum of mice we cannot discard larger differences between scrapie and control mice in homogeneous groups.

In the materials and methods section, a short explanation has been added in section 4.1.1 lines 558-563 and section 4.1.2 lines 581-582 to clarify that all animals had similar ages to exclude the effect of age on methylation changes. We have also included the correlations performed between the methylation parameters and age in the control groups (section 4.6 lines 720-722; results section 2.2.1. lines 155-158; section 2.3.1 lines 206-210 and lines 237-238; section 2.3.2. lines 259-261; Discussion section lines 370-377).

2. I also would like to see the scrapie characterization as a main figure. 

In the results section , section 2.1 “scrapie characterization” has been added. In Figure 1  PrPSc deposition and vacuolization scores of Tg338 mice are represented.

Regarding the scrapie characterization in sheep, these data have already been published in the paper “Betancor, M.; Pérez-Lázaro, S.; Otero, A.; Marín, B.; Martín-Burriel, I.; Blennow, K.; Badiola, J.J.; Zetterberg, H.; Bolea, R. Neurogranin and Neurofilament Light Chain as Preclinical Biomarkers in Scrapie. Int. J. Mol. Sci. 2022, 23, doi:10.3390/IJMS23137182/S1” (reference 34).

 3. Finally, when presenting relative expression for qPCR, one lose the exact level of expression. it would be good to see the CT values of the experiments presented in suppl material.

The CT values of the RT-qPCR analyses are presented in the Supplementary Tables S1 and S2, which have also been cited in the text: section 2.2.1. lines 153-154 and section 2.2.2. lines 186-188.

Reviewer 2 Report

The title of the manuscript is attractive and gathers the attention of readers. However, the paper describes 5mC and 5hmC levels assumed based on the immunocytochemical images processed by Image J software and as the authors mentioned in the discussion, further studies including bisulfite sequencing may be conducted in future. The overall scientific significance of the paper is average; there is no breakthrough discovery directly linking 5mC and 5hmC with the progression of prion diseases. However, I do consider that publishing this kind of scientific paper is important for the community. Data collected and published by the authors may be an inspiration for others or just serve as reference data to further investigate the link between DNA epigenetics and the pathogenesis and progression of prion diseases.

I suggest the authors check three minor issues:

  1. I recommend reviewing the names of animal groups (control, clinical, preclinical, clinical control, clinical mice etc.). Authors should make it consistent and perhaps should describe those groups in the "results" section (because the "methods" section is placed at the end of the paper, which makes it a bit difficult for readers to understand the exact meanings of those groups.
  2. Figures 3, 5, 7, and 9 are presented as line charts, but I consider them misleading because such charts are primarily used to visualize a trend in data over intervals of time, while on the x-axis there are "brain areas". I recommend authors reconsider the chart type for these figures.
  3. The authors use p-values 0.05 and 0.01 for different data within the work. I understand the intention of the authors to underline the higher significance of difference with p<0.01 over those with p<0.05, but I think that the statistical significance should be consistently established over the whole work, i.e. at 0.05.

I also suggest rephrasing "decrease of this biomolecule" (line 268), because the "biomolecule" refers to 5mC, while it is a residue within a biomolecule, namely, DNA. Perhaps a "decrease of this epigenetic modification" might be fine.

These are my comments on the manuscript, which I consider as prepared very well.

Author Response

We would like to thank this reviewer for the effort made to read and review this manuscript and for his/her comments that have contributed to improve it. You can find hereafter how we have addressed your suggestions. Reviewer comments are in bold and italic characters.

The title of the manuscript is attractive and gathers the attention of readers. However, the paper describes 5mC and 5hmC levels assumed based on the immunocytochemical images processed by Image J software and as the authors mentioned in the discussion, further studies including bisulfite sequencing may be conducted in future. The overall scientific significance of the paper is average; there is no breakthrough discovery directly linking 5mC and 5hmC with the progression of prion diseases. However, I do consider that publishing this kind of scientific paper is important for the community. Data collected and published by the authors may be an inspiration for others or just serve as reference data to further investigate the link between DNA epigenetics and the pathogenesis and progression of prion diseases.

We would like to thank this reviewer for his/her positive comments.

I suggest the authors check three minor issues:

  1. I recommend reviewing the names of animal groups (control, clinical, preclinical, clinical control, clinical mice etc.). Authors should make it consistent and perhaps should describe those groups in the "results" section (because the "methods" section is placed at the end of the paper, which makes it a bit difficult for readers to understand the exact meanings of those groups.

In the results sections 2.2.1 lines 143-145 and 2.2.2 lines 164-166 a brief explanation of the animal groups has been added for better understanding of the readers.

2. Figures 3, 5, 7, and 9 are presented as line charts, but I consider them misleading because such charts are primarily used to visualize a trend in data over intervals of time, while on the x-axis there are "brain areas". I recommend authors reconsider the chart type for these figures.

Thank you for your recommendation, we have used this type of graph for representing our immunohistochemical data although they are not a data trend to homogenize our results with those from other articles that used this type of chart to represent PrPSc deposition, prion related lesions and specific markers in different brain regions. Some examples can be found in the following papers :

- Vidal, E.; Fernández-Borges, N.; Pintado, B.; Eraña, H.; Ordóñez, M.; Márquez, M.; Chianini, F.; Fondevila, D.; Sánchez-Martín, M.A.; Andreoletti, O.; et al. Transgenic Mouse Bioassay: Evidence That Rabbits Are Susceptible to a Variety of Prion Isolates. PLoS Pathog. 2015, 11, doi:10.1371/JOURNAL.PPAT.1004977.

- Vidal, E.; Sánchez-Martín, M.A.; Eraña, H.; Lázaro, S.P.; Pérez-Castro, M.A.; Otero, A.; Charco, J.M.; Marín, B.; López-Moreno, R.; Díaz-Domínguez, C.M.; et al. Bona Fide Atypical Scrapie Faithfully Reproduced for the First Time in a Rodent Model. Acta Neuropathol. Commun. 2022, 10, doi:10.1186/S40478-022-01477-7.

- Betancor, M.; Pérez-Lázaro, S.; Otero, A.; Marín, B.; Martín-Burriel, I.; Blennow, K.; Badiola, J.J.; Zetterberg, H.; Bolea, R. Neurogranin and Neurofilament Light Chain as Preclinical Biomarkers in Scrapie. Int. J. Mol. Sci. 2022, 23, doi:10.3390/IJMS23137182/S1.

- Dustan, B.H.; Spencer, Y.I.; Casalone, C.; Brownlie, J.; Simmons, M.M. A Histopathologic and Immunohistochemical Review of Archived UK Caprine Scrapie Cases. Vet. Pathol. 2008, 45, 443–454, doi:10.1354/VP.45-4-443/ASSET/IMAGES/LARGE/10.1354_VP.45-4-443-FIG2.JPEG.

3. The authors use p-values 0.05 and 0.01 for different data within the work. I understand the intention of the authors to underline the higher significance of difference with p<0.01 over those with p<0.05, but I think that the statistical significance should be consistently established over the whole work, i.e. at 0.05.

In the materials and methods section, the statistical significance has been defined as P<0.05 in all the conducted experiments. In the results section, we indicate the different P-values using inequalities (P<0.05, p<0.01 and P<0.001) in the gene expression and immunohistochemical analyses, and provide the exact P-value in the correlation analyses to represent in a more precise way the P-values, as some journals recommend to give accurate P-values instead of only a global statistical significance of P<0.05 (Zhu 2016; Boos and Stefanski 2011).

- Boos, Dennis D., and Leonard A. Stefanski. 2011. “P-Value Precision and Reproducibility.” The American Statistician 65 (4): 213–21. https://doi.org/10.1198/TAS.2011.10129.

- Zhu, Weimo. 2016. “P <   0.05, < 0.01, < 0.001, < 0.0001, < 0.00001, < 0.000001, or < 0.0000001 ….” Journal of Sport and Health Science 5 (1): 77–79. https://doi.org/10.1016/J.JSHS.2016.01.019.

4. I also suggest rephrasing "decrease of this biomolecule" (line 268), because the "biomolecule" refers to 5mC, while it is a residue within a biomolecule, namely, DNA. Perhaps a "decrease of this epigenetic modification" might be fine.

Thank you for the comment. The sentence has been rephrased according to the reviewer suggestion: line 358 “a decrease of this epigenetic modification was reported”.

These are my comments on the manuscript, which I consider as prepared very well.

We thank this reviewer for his/her positive comments.

Reviewer 3 Report

In this paper, authors investigate the involvement of DNA methylations and epigenetic regulatory enzymes in the neuropathology of experimental and natural scrapie disease. This study follows a handful of papers claiming for a role of epigenetic mechanisms in the pathogenesis of prion disorders, in line with observations conducted in the last decade on other amyloidotic neurodegenerative disorders. Authors of this manuscript are not unaware of the field and design a promising complementary approach to get into the problem by studying brains from natural sheep scrapie and from mouse-adapted experimental scrapie. No much to comment about the work apart from its main point of weakness, represented by the absence of a clear correlation (either in sheep or in mice) between the phase of the disease, the distribution of lesions and the many biomarkers that have been investigated. In addition to that, alterations found in mice differ from those in sheep, making the whole picture even more difficult to understand. Clearly, we have to consider that these observations may be a typical trait of natural and experimental scrapie, yet, as it occurs with negative and heterogeneous results, it is utterly necessary to get further strong experimental evidences in order to achieve a full assessment of the matter. Owing to this, conclusions from this work, while worthy of publication, have to be considered only provisional as correctly stated in the final part of the manuscript.

Minor issue. Please quote this paper in the manuscript: E.A. Viré and S. Mead. (2022), Gene expression and epigenetic markers of prion diseases. Cell Tissue Res, doi 10.1007/s00441-022-03603-2

Author Response

We would like to thank this reviewer for the effort made to read and review this manuscript and for his/her comments that have contributed to improve it. You can find hereafter how we have addressed your suggestions. Reviewer comments are in bold and italic characters.

In this paper, authors investigate the involvement of DNA methylations and epigenetic regulatory enzymes in the neuropathology of experimental and natural scrapie disease. This study follows a handful of papers claiming for a role of epigenetic mechanisms in the pathogenesis of prion disorders, in line with observations conducted in the last decade on other amyloidotic neurodegenerative disorders. Authors of this manuscript are not unaware of the field and design a promising complementary approach to get into the problem by studying brains from natural sheep scrapie and from mouse-adapted experimental scrapie. No much to comment about the work apart from its main point of weakness, represented by the absence of a clear correlation (either in sheep or in mice) between the phase of the disease, the distribution of lesions and the many biomarkers that have been investigated. In addition to that, alterations found in mice differ from those in sheep, making the whole picture even more difficult to understand. Clearly, we have to consider that these observations may be a typical trait of natural and experimental scrapie, yet, as it occurs with negative and heterogeneous results, it is utterly necessary to get further strong experimental evidences in order to achieve a full assessment of the matter. Owing to this, conclusions from this work, while worthy of publication, have to be considered only provisional as correctly stated in the final part of the manuscript.

Minor issue. Please quote this paper in the manuscript: E.A. Viré and S. Mead. (2022), Gene expression and epigenetic markers of prion diseases. Cell Tissue Res, doi 10.1007/s00441-022-03603-2

The paper has been cited in the introduction section line 99 as the reference 33.